# In-Hospital Mortality Among Patients Undergoing Percutaneous Pericardiocentesis for Pericardial Effusion with and Without Malignancy

**DOI:** 10.3390/curroncol32090514

**Published:** 2025-09-15

**Authors:** Ju Young Bae, Dae Yong Park, Soumya Banna, Jiun-Ruey Hu, Amr Saleh, Mamas A. Mamas, Robert L. McNamara, Michael G. Nanna, John F. Setaro, Luke K. Kim, S. Elissa Altin

**Affiliations:** 1Division of Cardiovascular Medicine, Weill Cornell Medicine, New York Presbyterian Hospital, New York, NY 10065, USA; afc9009@nyp.org (J.Y.B.);; 2Department of Internal Medicine, Section of Cardiovascular Medicine, Yale New Haven Hospital, New Haven, CT 06510, USA; 3Department of Internal Medicine, Section of Cardiovascular Medicine, University of California San Francisco, San Francisco, CA 94115, USA; 4Department of Cardiology, Smidt Heart Institute, Cedars-Sinai Medical Center, Los Angeles, CA 90048, USA; 5Department of Internal Medicine, St. Mary’s Hospital, Trinity Health of New England, Waterbury, CT 06706, USA; 6Keele Cardiovascular Research Group, Keele University, Newcastle ST5 5BG, UK; 7Division of Cardiology, West Haven VA Medical Center, West Haven, CT 06516, USA

**Keywords:** malignancy, pericardial effusion, pericardial tamponade, pericardiocentesis

## Abstract

Pericardial effusion, or fluid buildup around the heart, can result from both cancer-related and non-cancer-related causes. When severe, it often requires drainage through a procedure called pericardiocentesis. However, differences in outcomes between these two groups have not been clearly defined. Using a large national database, we compared outcomes in patients with and without cancer who underwent pericardiocentesis. We found that patients with cancer-related effusions had a higher in-hospital mortality, were more likely to be discharged to a care facility rather than home, and had slightly longer hospital stays with increased healthcare costs. These findings suggest that cancer-related pericardial effusions represent a more severe clinical condition and underscore the importance of individualized, goal-directed care in this high-risk population.

## 1. Introduction

Malignancy is a leading cause of pericardial effusion, accounting for roughly 30% of all symptomatic effusions [1]. Despite contemporary advances in oncologic therapies, malignant pericardial effusions (MPEs) are still associated with high rates of morbidity and mortality, reported between 44 and 56.3% [2,3,4]. Although pericardiocentesis is considered a relatively safe diagnostic and therapeutic modality for pericardial effusion [5], procedural intervention and hospitalization may not be within the goals of care if the prognosis is known and candidly discussed. As a result, the management of MPE is often nuanced, involving balancing the benefits and risks of procedural intervention in the context of a potentially limited life expectancy.

Despite the high prevalence of MPEs, little is known about the differences in post-pericardiocentesis mortality compared to patients with non-malignant pericardial effusions (NMPEs). This study aims to describe the differences in patient characteristics and in-hospital outcomes following pericardiocentesis in patients with an MPE and NMPE, using a nationally representative database in a contemporary cohort in order to obtain a better understanding of the prognosis for these patients when considering whether to perform an invasive procedure.

## 2. Method

### 2.1. Data Source

This is a retrospective cohort study utilizing the National Inpatient Sample (NIS), a clinical administrative database produced by the collaborative effort of the Healthcare Cost and Utilization Project (HCUP) and the Agency for Healthcare Research and Quality [6]. The database samples about 20% of hospitals across 49 participating states in the United States to represent all inpatient hospitalizations, close to 35 million occurrences per year, in the nation. Each admission is weighted to reflect national estimates, allowing for the accurate representation of overall admissions.

The purpose of the database is to monitor the occurrence of specific disease entities and their associated healthcare costs, outcomes, and quality measures. NIS adheres to the Health Insurance Portability and Accountability Act Safe Harbor Provision by removing all state, hospital, and patient identifiers to guarantee patient confidentiality. This data is de-identified and publicly available through HCUP [6], and therefore exempt from institutional review board evaluation.

### 2.2. Study Population and Covariates

We identified all hospitalizations between 1 January 2016 and 31 December 2020 in which pericardiocentesis was performed. Pericardiocentesis was identified using the International Classification of Diseases, 10th Revision, Procedure Coding System (ICD-10-PCS). Hospitalizations were excluded if they involved patients under 18 years of age or if the hospital encounter was missing data for demographics, hospital characteristics, primary payer, median income, and in-hospital outcomes. In order to eliminate cases of cardiac tamponade that occurred as a result of instrumentation from cardiac procedures and cardiothoracic surgeries, hospitalizations were also excluded if the patient underwent cardiac catheterization, cardiac surgery (coronary artery bypass graft (CABG), surgical aortic valve replacement, surgical mitral valve replacement, mitral valve repair, surgical tricuspid valve replacement, tricuspid valve repair, and surgical pulmonary valve replacement), percutaneous coronary intervention (PCI), device placement, or endomyocardial biopsy during that admission (*n* = 15,745).

Demographics, hospital characteristics (region, bed size, and urban location), primary payer, and ZIP code-based median income were extracted. A range of comorbidities and relevant medical conditions associated with pericardial effusion were selected (Appendix A). The cohort was stratified by the presence of malignancy, which was classified into 22 different primary types of cancer (oropharyngeal, esophageal, gastric, colorectal, anal, hepatobiliary, pancreatic, lung, skin, breast, cervical, uterine, ovarian, prostate, renal, bladder, central nervous system, thyroid, Hodgkin lymphoma, non-Hodgkin lymphoma, multiple myeloma, and leukemia). Benign and in situ neoplasms were not included. Comorbidities and diagnoses were determined using International Classification of Diseases, 10th Revision, Clinical Modification (ICD-10-CM) codes. Procedures were determined using ICD-10-PCS codes. All the codes used in our study can be found in the Appendix A.

### 2.3. Study Outcomes

The primary outcome was in-hospital mortality. Secondary outcomes included non-home discharges, length of stay, and total hospital cost. The total hospital cost was approximated by multiplying the total hospital charge, available in the original NIS databases, with the cost-to-charge ratios available in a separate website by HCUP [7].

### 2.4. Statistical Analysis

Survey analysis methods were based on weights given to hospital encounters, which were used to generate results representative of national estimates. When describing the main characteristics, categorical variables were shown as percentages, and continuous variables were shown as means with confidence intervals. When comparing categorical variables, the chi-squared test was used, and when comparing continuous variables, Student’s *t*-test was used. For both primary and secondary outcomes, which are binary, logistic regression was used to calculate odds ratios with respective 95% confidence intervals (CI). For secondary outcomes, which are continuous, linear regression was used to calculate the mean difference with its 95% CI.

Multivariable models were adjusted using the following covariates: age, sex, hypertension, diabetes mellitus, congestive heart failure, chronic ischemic heart disease, atrial fibrillation, chronic obstructive pulmonary disease, liver cirrhosis, anemia, malnutrition, thrombocytopenia, and coagulopathy. The adjusted covariates were chosen for their clinical relevance and significance after a consensus was achieved from multiple team discussions. A separate analysis, which examined covariates associated with in-hospital mortality using the same cohort, was also conducted. A total of 53 covariates (age, sex, types of cancer, relevant medical conditions, and comorbidities) were included in a multivariable logistic regression model, and Bonferroni correction (*p*-value < 0.0009 for statistical significance) was applied to select covariates associated with higher odds of in-hospital mortality. Aside from this analysis, all tests were 2-sided, and *p*-values < 0.05 were considered significant given the exploratory and not confirmatory nature of our observational study. Data curation and all statistical analyses were performed using SAS, version 9.4 (SAS Institute, Cary, NC, USA).

## 3. Results

### 3.1. Baseline Demographics

A total of 85,125 hospitalizations involving pericardiocentesis between 1 January 2016 and 31 December 2020 were identified. Of these, 24,740 admissions had a concomitant diagnosis of malignancy, while 60,385 did not (Figure 1). Baseline characteristics for each group are presented in Table 1. Patients with MPEs were generally younger and more likely to have a history of malnutrition, prior radiation therapy, palliative care interventions, and do-not-resuscitate orders. They were less likely to have hypothyroidism, end-stage renal disease (ESRD), acute pericarditis, autoimmune disease, or congestive heart failure (CHF) compared to those with NMPEs (Table 1, *p* < 0.001). Lung cancer was the most commonly associated malignancy, accounting for 40.3% of cases (Figure 2).

### 3.2. Mortality

The in-hospital mortality rates following pericardiocentesis were 11.8% in patients with malignancy and 8.2% in patients without (Table 2). In those with underlying malignancy, the unadjusted in-hospital mortality was highest for patients with effusion associated with hepatobiliary cancer (16.4%), cervical cancer (15.8%), esophageal cancer (14.6%), anal cancer (14.3%), and ovarian cancer (14.1%). Crude in-hospital mortality rates following pericardiocentesis in various malignancy subtypes are shown in Figure 3.

After multivariable adjustment, encounters for pericardiocentesis in patients with underlying malignancy was associated with higher odds (OR 1.50, CI:1.34–1.68, and *p* < 0.001) of in-hospital mortality compared with those without malignancy (Table 2). Among the different types of malignancy, lung cancer, non-Hodgkin lymphoma, leukemia, esophageal cancer, and ovarian cancer were associated with the increased adjusted odds of in-hospital mortality of 1.64 (CI:1.39–1.93, *p* < 0.001), 1.73% (CI:1.25–2.39, *p* < 0.001), 1.47 (CI:1.06–1.93, *p* < 0.001), 2.28 (CI: 1.25–4.16, *p* < 0.001), and 2.17 (CI:1.07–4.39, *p* < 0.001), respectively, when compared with patients with non-malignant effusions (Figure 4).

### 3.3. Secondary Outcomes

Patients with an MPE were more likely to have a non-home discharge compared to those with an NMPE. The length of stay (LOS 9.4 vs. 9.1 days; *p* = 0.001) and total hospital cost (USD 34,057 vs. USD 33,404; *p* < 0.001) were also marginally greater in those with an MPE (Table 2).

### 3.4. Covariates Associated with In-Hospital Mortality

Of the 53 covariates included in the model, only chronic obstructive pulmonary disease (CI: 1.21–1.89, *p* = 0.0003), malnutrition (CI: 1.47–2.24, *p* < 0.0001), and coagulopathy (CI: 1.86–3.44, *p* < 0.0001) were each associated with statistically higher odds of in-hospital mortality after the Bonferroni correction. None of the individual types of cancers were associated with significantly higher odds of in-hospital mortality after the Bonferroni correction in this model.

## 4. Discussion

In this nationally representative, retrospective analysis of hospitalizations for pericardiocentesis in those with and without malignancy, we report several key findings. First, those with any type of malignancy had a significantly higher unadjusted in-hospital mortality rate compared to those without malignancy. Second, after a multivariate adjustment, lung cancer, non-Hodgkin lymphoma, esophageal cancer, ovarian cancer, and leukemia were associated with a significantly increased risk of death during the same admission. Lastly, patients with MPEs were more likely to have marginally greater hospital costs and length-of-stays compared to those with NMPEs. This is one of the few large-scale, contemporary analyses comparing in-hospital mortality and other key hospital outcomes among patients with and without malignancy undergoing pericardiocentesis.

This study corroborates previous findings that report lung (40%), breast (8%), and hematologic malignancies (15%) as the most prevalent cancer subtypes among patients with MPEs [5,8,9,10]. In cancer patients, the presence of MPEs suggests advanced disease progression and is associated with poor long-term survival [11]. However, data on short-term survival following pericardiocentesis are controversial. In the study by Matetic et al., the reported in-hospital mortality among patients with malignant pericardial effusion (MPE) undergoing pericardiocentesis was 15.6%, which is comparable to the in-hospital mortality rate of 11.6% observed in our cohort [8]. This contrasts with the mortality reported in most of the studies investigating in-hospital mortality among patients with MPEs. For instance, a single-center study of 502 patients undergoing pericardiocentesis reported an in-hospital mortality rate of 46.9% for those with malignancy, compared to 14.7% for those without malignancy (46.9% vs. 14.7%; *p* < 0.001) [4]. This is significantly higher than the mortality rate reported in our study and may partially be explained by a smaller sample size. Conversely, a larger National Inpatient Sample (NIS)-based analysis of cancer patients treated during an earlier time period (between 2004 and 2017) reported an in-hospital mortality rate of 15.6% among those with a presumed MPE who underwent pericardiocentesis, which aligns closely with our findings [8]. While previous studies have investigated mortality in patients with various types of malignancies based on the presence or absence of pericardial effusion, our study contributes to the existing literature by specifically comparing in-hospital mortality rates following pericardiocentesis in patients with malignant versus non-malignant pericardial effusions. This distinction is crucial, given the prognostic implications of MPEs and how these may influence treatment strategies and patient outcomes.

In addition, in this analysis, lung cancer, non-Hodgkin’s lymphoma, and leukemia were associated with significantly increased odds of in-hospital death following pericardiocentesis. This is consistent with the existing literature reporting that MPE in those with lung cancer is associated with a significantly shorter survival [12,13]. For hematologic malignancies, however, our findings are contrary to the previously published data that suggested MPEs associated with lymphoma or leukemia were associated with better survival than those with solid tumors [5,14,15]. In the current study, leukemia and non-Hodgkin lymphoma were associated with increased in-hospital mortality (47% and 73%), respectively, during admissions involving pericardiocentesis; however, the underlying mechanisms for this association remain unclear. It may be speculated that baseline patient characteristics—such as comorbidities, the type of cancer treatment, time of diagnosis, or disease stage—could play a role. The latter factors are not available to us for analysis in the current database. Additionally, MPEs related to breast cancer accounted for approximately 8% of pericardiocentesis cases, with an in-hospital mortality rate of just 4%. This suggests that while breast cancer-related MPE is common, it appears to have a less pronounced association with mortality outcomes compared to other malignancies [16]. Furthermore, esophageal and ovarian cancers were shown to have the greatest adjusted odds of in-hospital mortality of 2.28 (CI: 1.25–4.16, *p* < 0.001) and 2.17 (CI:1.07–4.39, *p* < 0.001), respectively. Although a malignant pericardial effusion secondary to ovarian cancer is rare, the elevated mortality may reflect delayed recognition due to its atypical presentation, further compounded by the fact that ovarian cancer is frequently diagnosed at an advanced, metastatic stage. Similarly, esophageal cancer is often identified at a late stage, which may contribute to the higher observed in-hospital mortality in these patients [17].

Lastly, this study found a marginally greater hospital length of stay and cost among patients undergoing pericardiocentesis with an MPE compared to NMPE. Long and repeat hospital admissions toward the end of life have previously been established as markers of a poor quality of life [18]. Recent data suggest 1 in 5 patients with MPEs who undergo pericardiocentesis develop re-accumulation of fluid within 30 days [19], with a higher 30-day cumulative cost compared to surgical drainage [20]. Further studies utilizing the NRD may provide longitudinal insights about the cumulative time of hospitalization for the management of MPEs and subsequent costs. In a recent study, the inaccurate assessment of the prognosis was shown to be a key driver of hospitalizations and inpatient costs near the end of life [21,22]. Taken together, this highlights the importance of honest goals of care discussions about post-procedure expectations and the overall cancer prognosis once patients with certain malignancies present with pericardial effusion. With the emerging role of immunotherapy and immune checkpoint inhibitors in the treatment of various malignancies, it is important to recognize potential cardiotoxicities, including pericardial disease. Since the mainstay of management is medical therapy and the discontinuation of the offending agents and accurate recognition and differentiation of the underlying etiology are crucial for guiding prognosis and reducing mortality [23,24,25].

## 5. Limitations

These results should be interpreted in the context of several limitations. First, this database does not have specific information pertaining to cancer stage, metastatic status, cause of death or treatment details. Second, the NIS has no data patient-centered outcomes, such as quality of life, which is a key consideration in this population. Third, there are coding issues inherent to using a large, national administrative database. Fourth, the NIS does not link readmissions or repeat procedures, which may be common among those with pericardial effusions. Lastly, there were notable differences between groups; residual confounding is possible.

## 6. Conclusions

In conclusion, this study demonstrates that patients with an MPE requiring pericardiocentesis had significantly higher inpatient mortality compared to patients with an NMPE requiring pericardiocentesis, specifically in those with lung cancer and hematologic malignancies. Future studies are needed to understand risk factors and predictors of higher in-hospital mortality, which may help to identify patients who may benefit from more conservative management, if it is consistent with patients’ stated goals of care.

## Figures and Tables

**Figure 1 curroncol-32-00514-f001:**
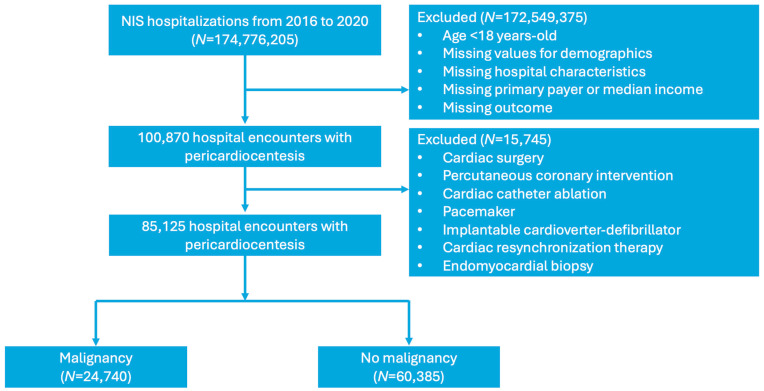
Flow chart detailing the selection of hospitalizations between 2016 and 2020 requiring pericardiocentesis in patients with and without malignancy.

**Figure 2 curroncol-32-00514-f002:**
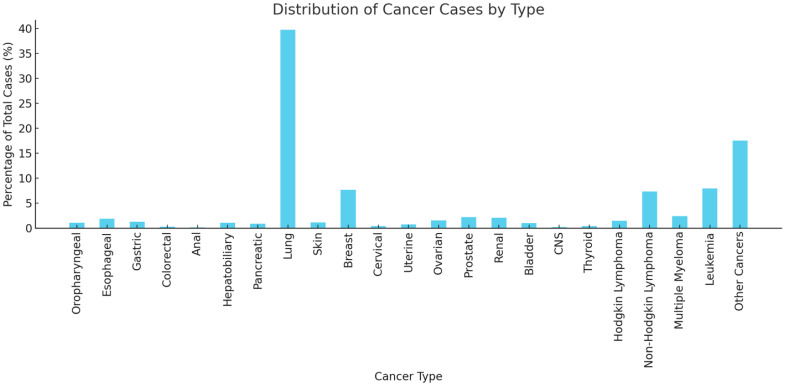
Distribution of cancer cases undergoing pericardiocentesis by type.

**Figure 3 curroncol-32-00514-f003:**
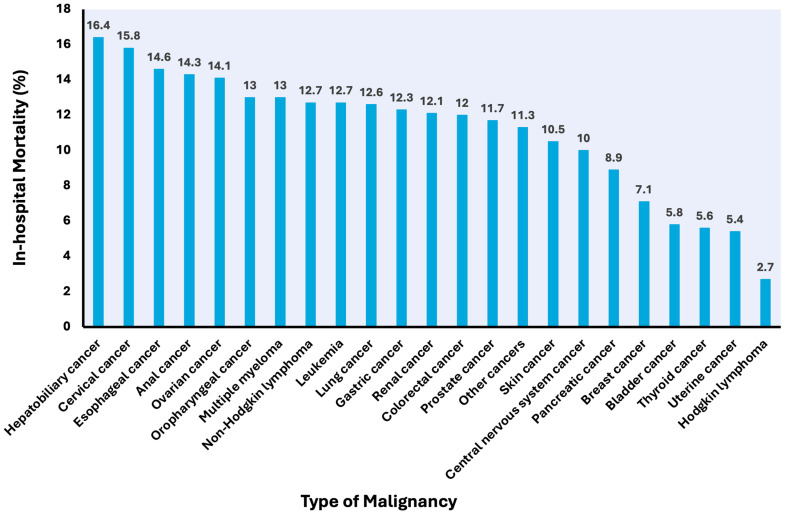
In-hospital mortality after pericardiocentesis for malignant pericardial effusion in different types of malignancy.

**Figure 4 curroncol-32-00514-f004:**
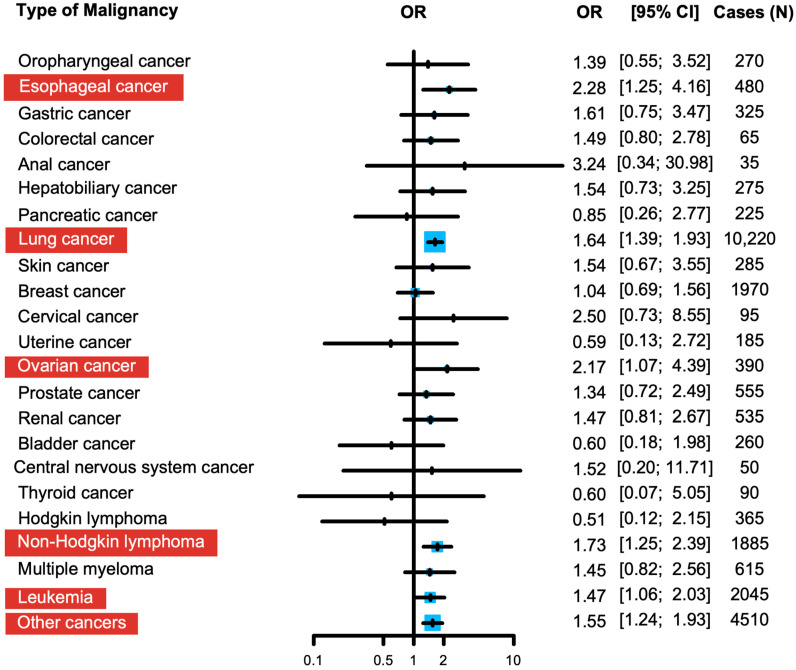
Adjusted in-hospital mortality in different types of malignant pericardial effusion versus non-malignant pericardial effusion. Odds ratios (OR) and 95% confidence intervals (CI) are displayed both visually and numerically. Cancers with statistically significant odds ratios for in-hospital mortality are highlighted in red. The size of the blue square is proportional to the sample size for that cancer.

**Table 1 curroncol-32-00514-t001:** Main characteristics of hospitalizations where pericardiocentesis was performed.

	Malignancy (+)	Malignancy (−)	*p*-Value
**Number of hospitalizations**	24,740	60,385	<0.001
**Male sex (%)**	49.3	50.9	0.053
**Age, mean (SE)**	62.1 (14.2)	62.8 (17.1)	0.005
**Race (%)**			<0.001
White	70.1	67.3	
Black	12.8	16.0	
Hispanic	8.1	10.5	
Asian	5.1	2.9	
AI/AN	0.4	0.4	
Other	3.5	2.9	
**Relevant medical conditions (%)**			
Hypothyroidism	12.9	15.4	<0.001
End-stage renal disease	2.6	11.4	<0.001
Acute pericarditis	7.5	14.4	<0.001
Infective endocarditis	0.3	1.0	<0.001
History of irradiation	11.1	2.1	<0.001
Systemic lupus erythematosus	0.5	3.1	<0.001
Rheumatoid arthritis	1.7	3.6	<0.001
Systemic sclerosis	0.3	0.9	<0.001
Inflammatory myopathy	0.0	0.1	0.292
Sjogren’s disease	0.2	0.5	0.037
Sarcoidosis	0.2	0.6	<0.001
**Comorbidities (%)**			
Smoking	45.6	34.9	<0.001
Hypertension	30.8	27.4	<0.001
Diabetes mellitus	19.4	31.0	<0.001
Hyperlipidemia	31.1	43.3	<0.001
Obesity	9.5	20.7	<0.001
Congestive heart failure	19.2	38.1	<.0001
Chronic ischemic heart disease	14.2	18.8	<0.001
Atrial fibrillation	30.0	37.9	<0.001
COPD	23.9	16.2	<0.001
Pulmonary hypertension	4.0	9.4	<0.001
Chronic kidney disease	15.2	31.9	<0.001
Liver cirrhosis	3.1	4.3	<0.001
Dementia	1.8	3.5	<0.001
Anemia	34.0	34.0	0.974
Malnutrition	18.8	9.3	<0.001
Coagulopathy	13.3	13.1	0.776
Thrombocytopenia	7.7	6.6	0.011
Palliative care consult (%)	16.6	3.7	<0.001
Pericardial tamponade (%)	55.8	44.5	<0.001
Surgical pericardial drainage (%)	8.8	6.2	<0.001
Mechanical ventilation (%)	11.8	13.8	<0.001
Need for vasopressor support (%)	4.4	4.5	0.825
**Hospital characteristics (%)**			
**Hospital region**			0.064
Northwest	22.8	21.1	
Midwest	23.2	22.5	
South	34.1	36.2	
West	19.9	20.2	
**Hospital bed size**			<0.001
Small	10.5	11.8	
Medium	23.1	25.5	
Large	66.4	62.7	
**Urban location**			0.068
Rural	2.4	2.9	
Urban non-teaching	12.8	13.6	
Urban teaching	84.8	83.6	
**Primary payer (%)**			<0.001
Medicare	46.4	56.3	
Medicaid	14.2	12.1	
Private insurance	34.5	25.9	
Self-pay	2.5	3.2	
No charge	0.2	0.3	
Others	2.2	2.2	
**Median income (%)**			0.001
Quartile 1	25.2	27.5	
Quartile 2	24.5	25.3	
Quartile 3	24.2	24.3	
Quartile 4	26.1	22.8	

**Table 2 curroncol-32-00514-t002:** Comparison of outcomes after pericardiocentesis for pericardial effusion with and without malignancy.

Outcome	Malignancy (+)	Malignancy (−)	Crude Odds Ratio	*p*-Value	Adjusted Odds Ratio	*p*-Value
In-hospital mortality (%)	11.8	8.2	1.49 (1.34–1.66)	<0.001	1.50 (1.34–1.68)	<0.001
Non-home discharge (%)	53.1	45.7	1.34 (1.26–1.44)	<0.001	1.40 (1.30–1.51)	<0.001
Length of stay (days ± SE)	9.6 ± 10.4	9.1 ± 11.5	0.56 (0.19–0.93) ^a^	0.001	0.43 (0.07–0.79)	0.019
Total hospital cost ($ ± SE)	34,057 ± 46,335	33,404 ± 51,955	654 ((−1013)–2321) ^a^	<0.001	546 ((−1088)–2180)	0.512

^a^ Absolute mean difference with 95% confidence intervals. Abbreviation: SE, standard error.

## Data Availability

The data that support the findings of this study are available on request from the corresponding author.

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
