# Peer review of "In-Hospital Mortality Among Patients Undergoing Percutaneous Pericardiocentesis for Pericardial Effusion with and Without Malignancy"

_curroncol, 2025, doi:10.3390/curroncol32090514_

Round 1
Reviewer 1 Report
Comments and Suggestions for Authors
Bae et al present an interesting work on
Esophageal and ovarian cancer were associated with greater adjusted odds of in-hospital mortality than lung etc (Fig 4). These types of Ca have not been discussed and certainly ignored in the abstract.
More references have to be added.
Which are the limitations of the study.
Authors are suggested to make a graph abstract after correcting the above flows
Author Response
Reviewer 1
Comment 1: “Esophageal and ovarian cancer were associated with greater adjusted odds of in-hospital mortality than lung etc. (Figure 4). These types of cancers have not been discussed and certainly ignored in the abstract.”
Response 1:
Thank you very much for this feedback. We agree that the increased association between esophageal and ovarian cancer with in-hospital mortality should be highlighted in the abstract. We have thus added to the Abstract:
“Lung cancer, non-Hodgkin lymphoma, esophageal cancer, ovarian cancer, and leukemia were associated with a significantly increased risk of death during the same admission” (abstract lines 47-49).
We also added to the Discussion:
“Esophageal and ovarian cancers were associated with the highest adjusted odds of in-hospital mortality, at 2.28 (95% CI: 1.25–4.16; P < 0.001) and 2.17 (95% CI: 1.07–4.39; P < 0.001), respectively. Although malignant pericardial effusion secondary to ovarian cancer is rare, the elevated mortality may reflect delayed recognition due to its atypical presentation, further compounded by the fact that ovarian cancer is frequently diagnosed at an advanced, metastatic stage. Similarly, esophageal cancer is often identified at a late stage, which may contribute to the higher observed in-hospital mortality in these patients.”
Comment 2: “More references have to be added.”
Response 2:
Thank you for this comment. More references have been added to the manuscript.
References added:
(17) Flores Castro JA, Rasha A, Pandu A, Mushiyev S. Cardiac Tamponade as the Initial Presentation of Metastatic Esophageal Adenocarcinoma. Cureus. 2021;13(8):e16863. Published 2021 Aug 3. doi:10.7759/cureus.16863
- Mudra, S.E.; Rayes, D.L.; Agrawal, A.; Kumar A.K.; Li J.Z.; Njus M.; McGowan K.; Kalam K.A.; Charalampous C.; Schleicher M.; et al. Immune checkpoint inhibitors and pericardial disease: a systematic review. Cardio-Oncol. 2024, 10, 29. https://doi.org/10.1186/s40959-024-00234-0
- Nardin, S.; Ruffilli, B.; Costantini, P.; Mollace, R.; Taglialatela, I.; Pagnesi, M.; Chiarito, M.; Soldato, D.; Cao, D.; Conte, B.; et al. Navigating Cardiotoxicity in Immune Checkpoint Inhibitors: From Diagnosis to Long-Term Management. Cardiovasc. Dev. Dis. 2025, 12, 270. https://doi.org/10.3390/jcdd12070270.
- Hu, J.R.; Florido, R.; Lipson, E.J.; Naidoo, J.; Ardehali, R.; Tocchetti, C.G.; Lyon, A.R.; Padera, R.F.; Johnson, D.B.; Moslehi, J. Cardiovascular toxicities associated with immune checkpoint inhibitors. Res. 2019, 115, 854–868. https://doi.org/10.1093/cvr/cvz026
Comment 3:“Which are the limitations of the study?”
Response 3:
We have enumerated the following limitations of the study:
“These results should be interpreted in the context of several limitations. First, this database does not have specific information pertaining to cancer stage, metastatic status, cause of death or treatment details. Second, NIS has no data patient-centered outcomes, such as quality of life, which is a key consideration in this population. Third, there are coding issues inherent to using a using a large, national administrative database. Fourth, NIS, does not link readmissions or repeat procedures, which may be common among those with pericardial effusions. Lastly, there were notable differences between groups; residual confounding is possible.” (lines 269-276)
Comment 4: “Authors are suggested to make a graph abstract after correcting the above flows.”
Response 4: We have revised our graphical abstract to reflect the requests of both reviewers, as follows:
(please see the updated graphical abstract attached)

Reviewer 2 Report
Comments and Suggestions for Authors
This research is a cohort study of pericardiocentesis in patient with or without cancer. This manuscript utilizes a large population, registry-based outcomes. The statistical and data analysis methods applied are comprehensive and systematic.
I have a few minor comments attached.
Abstract: Well written, no concerns
Introduction: Well written, no concerns
Method: There is some confusion regarding PCI and CABG.
You write “Hospitalizations were also excluded if the patient underwent cardiac catheterization, cardiac surgery (coronary artery bypass graft (CABG)” but later in the methode you add “The performance of other relevant procedures, including PCI and CABG, was also recorded”.
Is there any information regarding the timing that may be lacking, or was later data/patients included for mortality but not for hospitalization?
Results. Well written,
Figure 4: The CI is large, so including the number of patients for all types of malignancy is recommended.
Discussion: well written.
Although this material does not include information regarding treatment. As the use of immunotherapy, particularly in lung cancer, continues to rise, it might be relevant to mention cardiovascular toxicity and pericardial effusion associated with immunotherapy and checkpoint inhibitors.
The supplementary material provided appears to be the same as the main manuscript (apart from figure 1) Is this the correct attachment?
Author Response
Reviewer 2:
Comment 1: “In the method section, you write hospitalizations were also excluded if the patient underwent cardiac catheterization, cardiac surgery, CABG, but later in the method you add the performance of other relevant procedures, including PCI and CABG was also recorded, Is there any information regarding that may be lacking or was later data/patients included for mortality but not for hospitalization?”
Response 1:
Thank you for pointing out this opportunity to improve the clarity of our Methods section. With regard to the sentence “The performance of other relevant procedures, including PCI and CABG, was also recorded”–procedure codes for PCI and CABG were identified solely for the purpose of excluding these hospitalizations. They were not used as downstream outcome measures. We have thus removed this sentence. We have also revised the Methods section to clarify:
“In order to eliminate cases of cardiac tamponade that occurred as a result of instrumentation from cardiac procedures and cardiothoracic surgeries, hospitalizations were also excluded if the patient underwent cardiac catheterization, cardiac surgery (coronary artery bypass graft (CABG), surgical aortic valve replacement, surgical mitral valve replacement, mitral valve repair, surgical tricuspid valve replacement, tricuspid valve repair, and surgical pulmonary valve replacement), percutaneous coronary intervention (PCI), device placement, or endomyocardial biopsy during that admission (n=15,745).” (lines 99-106)
Comment 2: Figure 4 : The CI is large, so including the number of patients for all types of malignancy is recommended
Response 2: We agree with the point that it would be helpful to see the number of cases to better understand the variation in confidence intervals. For example, in anal cancer, central nervous system cancer, and thyroid cancer, the case counts (35, 50, and 90, respectively) are concordant with the wide confidence intervals. We have made changes to Figure 4 to include the number of patients for each type of malignancy.
Comment 3: Although this material does not include information regarding treatment. As the use of immunotherapy, particularly in lung cancer, continues to rise, it might be relevant to mention cardiovascular toxicity and pericardial effusion associated with immunotherapy and checkpoint inhibitors.
Response 3: We agree that the growing use of immunotherapy, particularly immune checkpoint inhibitors, has given rise to a new class of cardiovascular toxicities, including pericarditis and pericardial effusion, that were not previously appreciated. We have added the following paragraph. “With the emerging role of immunotherapy and immune checkpoint inhibitors in the treatment of various malignancies, it is important to recognize potential cardiotoxicities, including pericardial disease. Since the mainstay of management is medical therapy and discontinuation of the offending agents, accurate recognition and differentiation of the underlying etiology are crucial for guiding prognosis and reducing mortality” (lines 262-267)
Comment 4: The supplementary material provided appears to be the same as the main manuscript (Apart from figure 1. Is this the correct attachment?
Response 4:
Thank you for identifying this error. The supplementary material provided should have been the following:
Table S1. List of the ICD-10-CM codes used
*ICD-10-PCS codes
Abbreviations: ICD-10-CM, International Classification of Diseases, Tenth Revision, Clinical Modification; ICD-10-PCS, International Classification of Diseases, Tenth Revision, Procedure Coding System
Quality of Figures: Reviewer 2
- Figures and tables can be improved
Response 1: Thank you for this feedback. We have made updates to the figures and tables.
*Please see the attached file for the updated figures/tables

Round 2
Reviewer 1 Report
Comments and Suggestions for Authors
Authors addressed all my concerns. The mscr can be published